# The Hindu Kush slab break-off as revealed by deep structure and crustal deformation

Sofia-Katerina Kufner [1,2,7 ✉], Najibullah Kakar [3,4], Maximiliano Bezada [2], Wasja Bloch [1], Sabrina Metzger [1], Xiaohui Yuan [1], James Mechie [1], Lothar Ratschbacher[5], Shokhruhk Murodkulov[6], Zhiguo Deng[1] & Bernd Schurr [1]

Break-off of part of the down-going plate during continental collision occurs due to tensile stresses built-up between the deep and shallow slab, for which buoyancy is increased because of continental-crust subduction. Break-off governs the subsequent orogenic evolution but real-time observations are rare as it happens over geologically short times. Here we present a finite-frequency tomography, based on jointly inverted local and remote earthquakes, for the Hindu Kush in Afghanistan, where slab break-off is ongoing. We interpret our results as crustal subduction on top of a northwards-subducting Indian lithospheric slab, whose penetration depth increases along-strike while thinning and steepening. This implies that break-off is propagating laterally and that the highest lithospheric stretching rates occur during the final pinching-off. In the Hindu Kush crust, earthquakes and geodetic data show a transition from focused to distributed deformation, which we relate to a variable degree of crust-mantle coupling presumably associated with break-off at depth.

[1] GFZ German Research Centre for Geosciences, Potsdam, Germany. [2] University of Minnesota, Minneapolis, MI, USA. [3] Norwegian Afghanistan Committee, Kabul, Afghanistan. [4] University of Potsdam, Potsdam, Germany. [5] TU Bergakademie Freiberg, Freiberg, Germany. [6] Tajik Academy of Sciences, Dushanbe, Tajikistan. [7] Present address: British Antarctic Survey, Cambridge, United Kingdom. ✉email: sofner@bas.ac.uk

At the closing of a plate tectonic cycle, subduction transitions to continental collision and subducted lithosphere detaches and is recycled into the mantle. This process has a strong influence on the magmatic, tectonic and basin-formation history of an orogen. Understanding the dynamics during the final detachment and linking these deep mantle processes to deformation in the crust are key aspects in understanding mountain-building processes[1,2]. The fraction and composition of the lithosphere returned into the mantle, in turn, influences the mantle chemical composition and contributes to global plate tectonics[3]. Yet, much of our understanding of slab detachment hinges on numerical modelling or on geological data of past events as the final pinching-off is supposed to happen ephemerally[4,5]. The Hindu Kush in Afghanistan, located at the western margin of the India-Asia collision zone (Fig. 1), is one of the few places where an ongoing detachment of lithosphere in a continental setting has been proposed, as frequent seismicity occurs at depths that are unusual for an intracontinental setting (Fig. 1a; 60–300 km depth—intermediate-depth seismicity). At the same time, these earthquakes offer the unique opportunity to study the deep structure of a mountain range through seismic tomography.

In body-wave tomography, the resolution largely depends on the number of crossing rays in the subsurface and the number and distribution of seismic stations within the study region. A good tool to image large-scale upper mantle structures is teleseismic tomography, which uses earthquakes located outside the study region. Different velocity models derived from teleseismic data exist for the Hindu Kush[6–10], consistently showing a high-velocity zone (HVZ) in the upper mantle. These anomalies have been interpreted either as a detaching slab of Indian origin[6–8,10,11] or as highly thickened, dense lithosphere, foundering drop-like into the more buoyant asthenosphere[9,12]. Each scenario imposes a distinctive shallow crustal deformation field[13] and may also influence the temperature and hence velocity structure of the crust in a characteristic way. However, because of the near-vertically incident rays at shallow levels, teleseismic data alone have a low resolution in the crust. Further, only large anomalies in the mantle can be resolved. Ref. [6] for instance, used a dense local seismic network (station spacing between 30 and

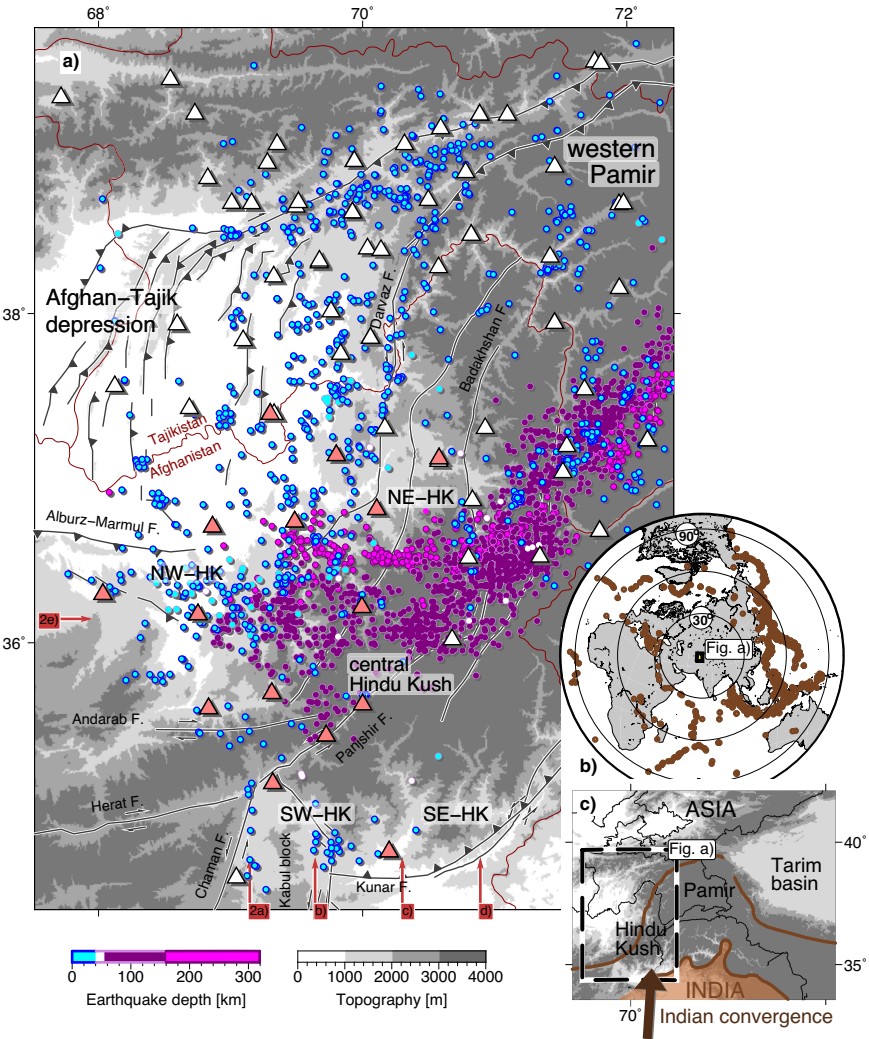

**Fig. 1 Geographic setting and data input for seismic tomography. a** Tectonic setting, local earthquakes used as inversion input, colour-coded by depth (circles), and seismic stations (triangles) with the campaign stations of the recent Afghanistan network (active 2017–2019) in light red. Faults (in black with white background) simplified after refs. [30,44,62] and references therein. Political boundaries in dark red. NE/NW/SE/SW-HK abbreviate north-east/north-west/south-east/south-west Hindu Kush. Starting points of profiles shown in Fig. 2 are highlighted in red. **b** Teleseismic earthquakes used as inversion input. **c** Study region in the context of the Indian-Asian collision zone where India converges at ~34 mm/year[63,64] northward towards Asia. The Cenozoic Indus-Yarlung suture (southern brown line) separates Indian from Asian rocks. The northern brown line represents the late Paleozoic-Triassic suture separating cratonic Asia in the north from the accreted Gondwana terranes in the south.

60 km), but still showed that a ~30 km thick anomaly at 200 km depth would be unresolvable using teleseismic data alone.

In the Hindu Kush, the crust above the mantle velocity anomalies comprises the Cenozoic Afghan-Tajik depression fold-thrust belt in the north and major strike-slip faults in the central Hindu Kush that dissect the crystalline basement rocks (Fig. 1a;[14–18]). The crust is highly thickened[19] and partly subducting to at least 150 km depth[20,21]. Neotectonic and geomorphological studies reveal regionally distributed tectonic activity[16,22] and positioning measurements of the global navigation satellite systems (GNSS)[23] show little horizontal shortening in the north-east Hindu Kush (NE-HK; Fig. 1a). The connection between this crustal deformation field and processes in the upper mantle is largely unknown.

Here, we present new data on crustal deformation and lithospheric-scale seismic images derived from finite-frequency coupled local and teleseismic P-wave tomography, which allow bridging interpretations on crustal vs. mantle deformation. Our approach combines local and teleseismic travel time data with their complementary depth coverage to obtain a well-resolved velocity image of both the crust and mantle. Resolution at depth is further sharpened through the use of sensitivity kernels at different frequency ranges rather than rays in the inversion[24–26]. We used data recorded by a recent, temporarily operated seismic network in Afghanistan, which covers for the first time the entire Hindu Kush region, overlying the deep mantle anomalies, as well as data from past deployments (Fig. 1a, Supplementary Table 1). Based on the tomograms, shallow earthquake fault plane solutions and new and existing GNSS data, we propose Indian subduction and ongoing slab break-off beneath the Hindu Kush and provide insight into coupling between deep processes and the crustal deformation field.

## Results

**Seismic imaging and slab model.** Our P-wave velocity model (Fig. 2) illuminates the crustal and mantle structure in the Hindu Kush, Afghan-Tajik depression and western Pamir from the surface to up to ~600 km depth. Details on the inversion strategy can be found in the 'Methods'. The mantle beneath the Hindu Kush is dominated by a prominent HVZ, which forms an east-striking, north-dipping slab (Fig. 2a–d). Its depth penetration, dip angle and thickness change along strike. The HVZ depth extent is shallowest (~350 km down-dip length), most gently dipping (~30° dip) and thickest (~200 km wide) in the west (Fig. 2a). It gradually penetrates deeper and thins in the upper 200 km towards the central Hindu Kush, where the HVZ ultimately reaches 600 km depth and dips nearly vertically or even appears gently overturned (Fig. 2d). The zone of intermediate-depth seismicity overlies the upper part of the HVZ, featuring a similar along-strike steepening and deepening towards the central Hindu Kush. The frequency of large magnitude earthquakes also increases towards the central Hindu Kush. All intermediate-depth earthquakes with magnitude >6.5 that struck the Hindu Kush in the last 30 years belong to the near-vertically dipping and highly thinned portion of the HVZ (Fig. 2c–e).

Compared to global averages[27], the Hindu Kush crust is characterized by a high-velocity upper crust (6.0–6.4 km/s; 0–5 km depth) and a domain of low-velocity middle/lower crust (crustal LVZ; 6.1–6.9 km/s; 30–60 km depth; Fig. 2 and Supplementary Fig. 2a). Beneath the central Hindu Kush, a mantle low-velocity zone (LVZ; 7.3–7.6 km/s; 80–160 km depth) extends this middle/lower crustal low-velocity domain down to ~160 km depth. The zone of intermediate-depth seismicity is sandwiched between the HVZ and the overlying LVZ (Fig. 2c, d). East of 71°E, we image a similar pair of high/low-velocity zones

and a deep earthquake zone, separated from the Hindu Kush anomalies (Fig. 2g–j). These anomalies have been attributed to the Pamir deep seismic zone[6,28,29] and will not be discussed in this paper.

**Resolution tests.** We conducted resolution tests to evaluate the performance of our combined local and teleseismic tomography approach (hereafter termed 'joint inversion') and to verify the robustness of the imaged velocity anomalies (see 'Methods' for details on the synthetic inversion procedure). First, the comparison of the joint inversion results for real and synthetic data with either only local or only teleseismic data shows that only the joint inversion can resolve both the crustal and mantle structure satisfactorily in terms of amplitude and geometry (Fig. 3a–c; further comparison in Supplementary Note 1), ultimately offering the least ambiguous basis for a tectonic interpretation.

Second, using the joint inversion approach, we tested different geometric configurations for the Hindu Kush mantle anomaly to discriminate between proposed processes responsible for the loss of mantle lithosphere (Fig. 3d). Subducted Asian lithosphere[23] and subducted Indian lithosphere[6–8,11] are represented by a south- and north-dipping anomaly, respectively (Fig. 3b, d-i). For the north-dipping scenario, we further tested different slab thicknesses (Fig. 3d-iii, d-iv). A vertical HVZ represents a hypothetical mantle drip[9,12] (Fig. 3d-ii). In addition, a model with a shallower slab termination and a neutral zone in the upper mantle is designed to evaluate whether vertical smearing strongly affects the inversion results (Fig. 4). All synthetic models feature the same crustal structure. Beneath the Hindu Kush, the crust consists of an upper crustal HVZ (0–15 km depth) and a middle/lower crustal LVZ. This pattern of velocity anomalies is inverted in the Afghan-Tajik depression crust (low-velocity sediments, high-velocity basement; see ref. [30] for detailed interpretation). In addition to the synthetic models of Figs. 3 and 4, a checkerboard test, which is a more generic proxy for ray coverage, is included in Supplementary Fig. 3.

The recovered anomalies in all synthetic test scenarios can be clearly distinguished from each other. However, only the north-dipping slab scenario, which includes a low-velocity zone overlying the slab to ~160 km depth (Fig. 3c) resembles the real data observations. Tests with different slab-thicknesses (Fig. 3d-iii, d-iv) illustrate that the thinning of the slab at 160–250 km depth is not a model artefact and that a slab of ~50 km thickness could still be resolved at upper mantle depths. The robustness of the W-E asymmetric slab-penetration depth is confirmed through the comparison shown in Fig. 4: a slab that terminates at 600 km depth along the entire W-E range covered here and a slab that terminates at ~360 km depth can be distinguished from each other. Thus, we can exclude that the varying depth penetration of the slab in the real data is an artefact of velocity smearing.

We note that the intensity of the recovered anomalies as well as the depth resolution varies locally (Fig. 4c), which is attributed to varying ray coverage and event distribution. Depth resolution is up to 600 km in the west (69°E) and east (71°E), whereas it is shallower in the centre (70°E). However, comparison of the synthetic models (Fig. 4-ii, iii) with the real data (Fig. 4-i) shows that all velocity anomalies introduced in the section above, specifically the tearing of the slab and the successive deepening of the slab, are within the resolution capacity of the tomographic inversion. Furthermore, global tomographies with larger resolution depth do not show a deeper penetration depth of the Hindu Kush slab either[7,8].

**Interpretation of the lithospheric structure.** Based on the real data inversion and synthetic tests, we conclude that the mantle

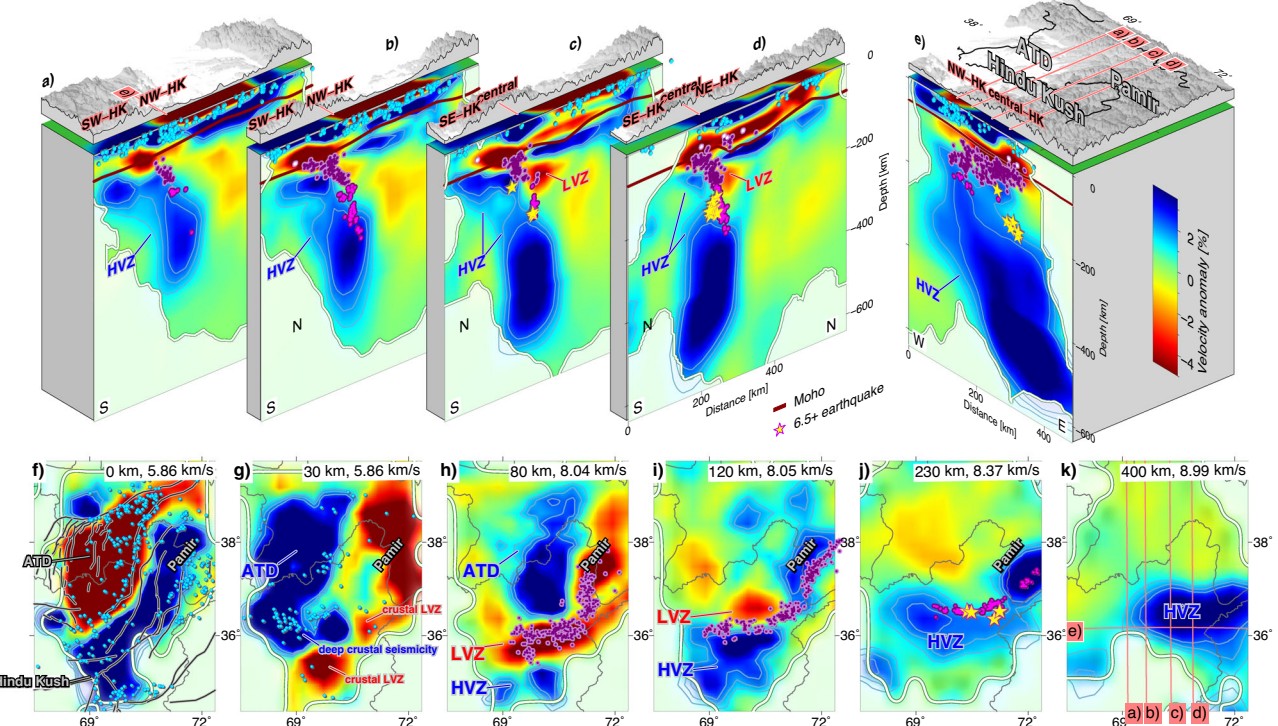

**Fig. 2 P-wave velocity tomographic model. a–e** Cross-sections with the topography on top; velocity anomaly is colour-coded by the percentage of variation relative to the initial 1D velocity model used in the inversion (Supplementary Fig. 1). The 1.0/1.5/2.0% velocity-anomaly contours are highlighted. The white line marks the resolution limit (see 'Methods'). The dark red line represents the crust-mantle boundary (Moho) constrained from receiver functions[19]. Local earthquakes within 15 km of the profile (circles) are colour-coded with depth as in Fig. 1a. Crustal events include those obtained from manual picking of the events recorded by the recent network in Afghanistan and those used in the inversion (see 'Methods'). Yellow stars represent magnitude 6.5+ earthquakes from USGS within the last 30 years projected from ±35 km swaths to account for the larger location uncertainties compared to the local catalogue. LVZ Hindu Kush low-velocity zone, HVZ Hindu Kush high-velocity zone, ATD Afghan-Tajik depression. Other abbreviations as in Fig. 1a. **f–k** Depth sections; section-depth and absolute P-wave velocity at this depth are given in the top right of each plot. Political boundaries in grey, faults in black with a white background as in Fig. 1a. Local earthquakes are plotted from ±5 km depth swaths in the crust and ±10 km depth swaths in the mantle. All other features as in panels (**a**)–(**e**).

HVZ beneath the Hindu Kush has a slab-like shape, dipping to the north, overlain by a region of focused intermediate-depth seismicity and an LVZ. The HVZ likely represents cold lithosphere subducted or foundered from shallower depths as velocity anomalies in the mantle are mostly due to temperature differences[31]. Together with the focused seismicity and the clear dip direction, this configuration agrees best with the scenario of a north-dipping Indian lithospheric slab beneath the Hindu Kush. The gradual eastward thinning, deepening and steepening of the subducted slab (Fig. 2a-e) then indicates the process of slab break-off[6,32–34], with the break-off being most advanced beneath the central Hindu Kush[20]. Such a configuration explains uniformly down-dip extensional focal mechanisms[6] and the accumulation of the largest earthquakes during the most advanced stage of break-off due to the largest strain rates occurring in the critically thinned portion of the slab[4]. Thus, the slab-penetration depth of ~600 km beneath the central Hindu Kush probably results from slab-stretching and break-off and does not represent the initial subduction length.

As the mantle LVZ overlies the lithospheric slab and geometrically connects with a region of over-thickened crust (as indicated by the seismic velocities resolved here and from receiver functions[19]), it likely represents crustal material pulled to mantle depths together with descending mantle lithosphere[20,21]. This requires coupling between the subducting crust and mantle lithosphere and explains why subduction of crustal material may be feasible despite its buoyancy[35]. The thickness of the LVZ is less

than the total crustal thickness (~20–30 km vs. 65 km;[19]), suggesting that only a part of the crust is pulled down. These are likely the lower and part of the middle crust as the upper crust has the lowest density and would more strongly resist subduction[4].

The LVZ, which we interpret as subducted crust terminates at ~160 km depth. Nevertheless, below ~160 km, subducted lower crust may be present, but eclogitized, making it indistinguishable from subducted mantle lithosphere[28]. In contrast, middle crust with the andesitic or granitic composition that undergoes high- or ultrahigh-pressure metamorphic mineral transitions retains its buoyancy to ~160 km depth[28,36] and hence should retain its low-density, low-velocity character (e.g., velocities below ~7.5 km/s[37]). Therefore, the middle crust is unlikely to subduct to depths greater than ~160 km. Just above the detachment point, the low velocities accumulate in a drop-like volume (Fig. 2c, d). This may represent ascending buoyant middle crust, possibly associated with melt generation while slab break-off is advancing[5,20]. In such a slab break-off and crustal-subduction scenario, the upper part of the intermediate-depth seismicity (~60–160 km; purple in Fig. 2) would mostly originate in the subducted crust as the earthquakes are geometrically located mostly within the LVZ overlying the mantle slab (Fig. 2). The earthquakes could then be due to phase transitions (i.e., eclogitization), which may lead to transformational faulting[38,39]. Isolated events may occur in the mantle lithosphere[40]. By contrast, the deepest, most vigorous seismicity occurs in high-velocity material (160–300 km; pink in

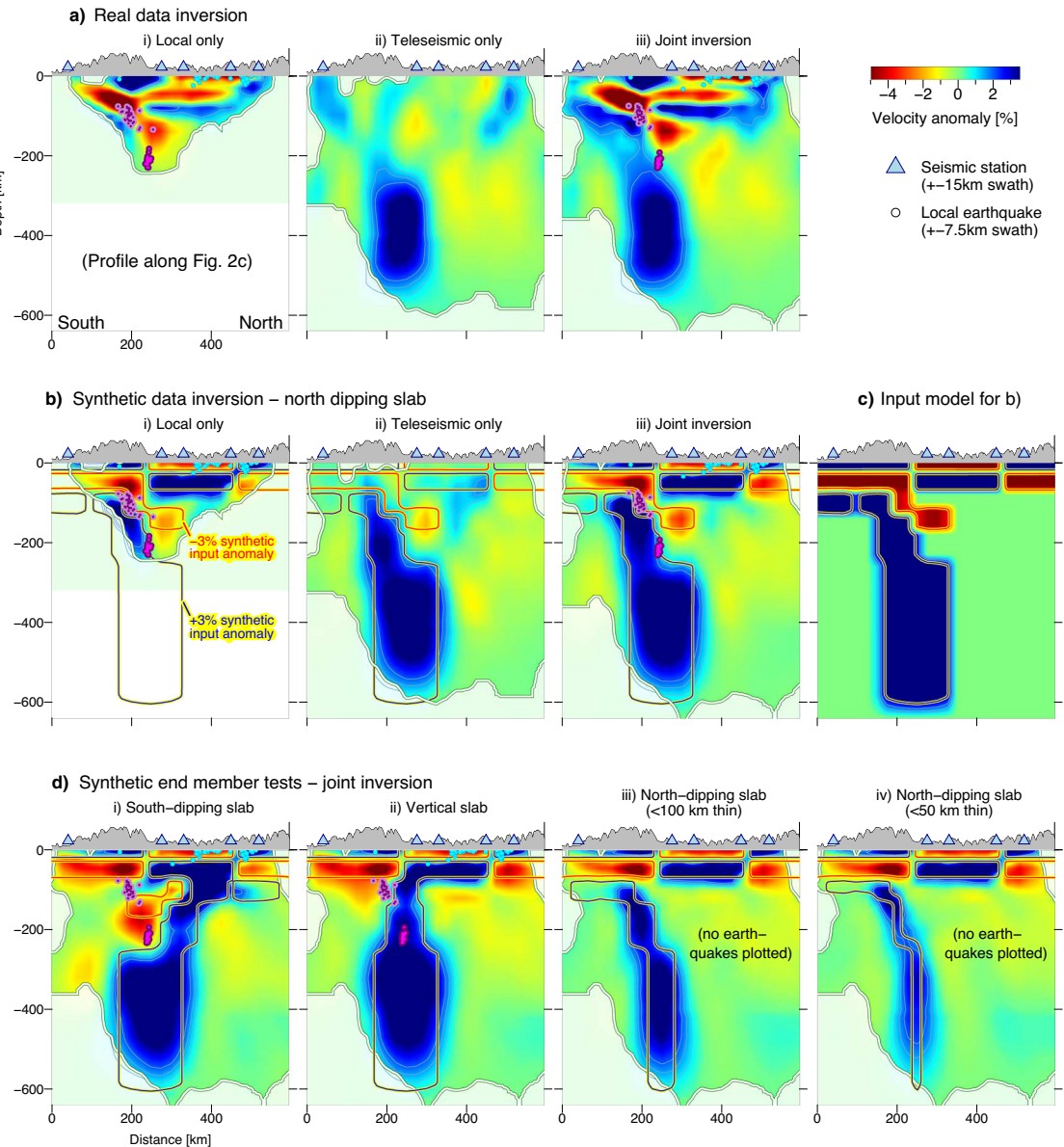

**Fig. 3 Data subset inversion and end member synthetic test for mantle anomaly geometry. a** Real data subset inversion for local only, teleseismic only and joint dataset. **b** Synthetic data subset inversion as in (**a**). The blue/red contours represent the ±3% velocity anomaly outline of the synthetic input model. **c** The synthetic input model used in (**b**), featuring a gradual increase of the velocity anomaly from 0 to 5%. **d** Synthetic models based on the combined dataset, testing different end-member scenarios for the geometry of the mantle anomaly. Local earthquakes, resolution limit and topography are plotted as in Fig. 2. All cross-sections are along with the profile of Fig. 2c.

Fig. 2), interpreted as mantle lithosphere. These earthquakes may result directly from slab break-off: under high strain rates[12,20,34] and relatively cold temperatures strain can localize along zones of reduced grain size due to shear-heating[41,42]. The resulting earthquakes indicate zones of active deformation.

**Crustal structure and deformation field.** To illuminate the relation between the slab break-off in the upper mantle and the crustal deformation field, we scrutinized all seismic data from the stations in Afghanistan to identify seismicity in the crust. This evaluation step was conducted manually and generated an event catalogue as complete as possible for our station deployment period (see details on crustal seismicity processing in the 'Methods'). It extends the event catalogue used for tomography, as the latter is subject to location quality restrictions and declustering (see details in the 'Methods').

We found that crustal seismicity shows a zonation in map view relative to the deep mantle slab (Fig. 5a, b). Crustal earthquakes in the central and north-eastern Hindu Kush (NE-HK in Fig. 5b) above the middle/lower crustal low-velocity domain and above the actively detaching part of the slab are sparse to absent. This observation seems to be consistent over longer observation periods as well (Supplementary Fig. 4). Interestingly, despite sparse crustal seismicity, GNSS rates indicate sinistral-transpressional displacement across this region (2.5 ± 1.8 mm/year sinistral displacement in the central Hindu Kush (GNSS1 in Figs. 5b) and 7.3 ± 1.0 mm/year in the NE-HK (GNSS2 in Fig. 5b); see 'Methods' for background on GNSS rates). This deformation is likely related to large-scale Indian northward motion, but displacement does not seem to be localized across one single fault. A broader area of deformation is also supported by neotectonic and geomorphological studies, which show tectonic activity to be

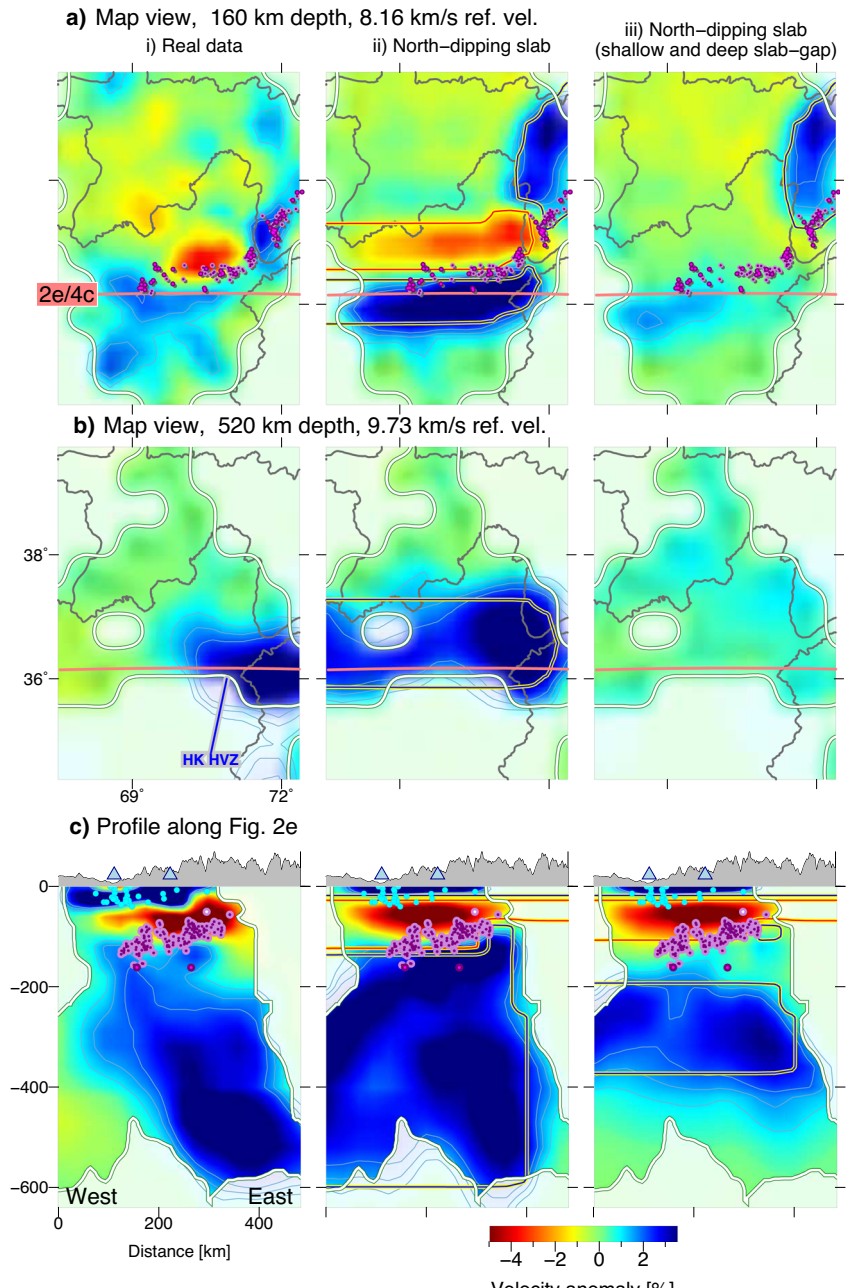

**Fig. 4 End member synthetic test for mantle anomaly resolution and depth extent. a, b** Map view sections. **c** Profile along Fig. 2e. **i** Real data inversion results as in Figs. 2 and 3a for comparison. **ii** North-dipping slab scenario as in Fig. 3b. **iii** A model with similar geometry to (ii) but no mantle high-velocity zone exists between 100 and 180 km depth and deeper than 360 km depth. Local earthquakes, resolution limit and topography are plotted as in Fig. 2.

distributed across the central Hindu Kush[16,22]. Together with the sparse crustal seismicity, this suggests that distributed, partly aseismic deformation may dominate in the central and NE-HK, overlying the middle/lower crustal low-velocity zone and the detaching part of the slab.

Outside the region of sparse seismicity, crustal earthquakes indicate an overall approximately NW-SE compressional stress regime (Fig. 5b and Supplementary Fig. 5) and cluster mostly along the transitions from low to high topography, i.e., in the north-western Hindu Kush along its margin to the Afghan-Tajik depression (NW-HK in Fig. 5b) and along the southern margin of the eastern Hindu Kush (SE-HK in Fig. 5b). Other clusters occur along strike-slip faults, in particular, the faults rimming the Kabul block (SW-HK in Fig. 5b) and west of the Badakhshan

fault. Domains of prevailing thrusting in the north-western (NW-HK) and south-eastern (SE-HK) foothills of the Hindu Kush are separated by a corridor of sinistral strike-slip deformation, grossly connecting the Chaman fault zone (SW-HK) in the south with the strike-slip faults of the NE-HK in the north. These faults likely accommodate the relative movements between the Pamir and Hindu Kush[43]. Crustal earthquakes are dominantly shallower than 10 km, which is the typical cut-off depth observed in adjacent crustal blocks[30,44]. An exception is a cluster of deeper events (up to 30 km depth) occurring at the southern margin of the Afghan-Tajik depression (NW-HK in Fig. 5b). Strikingly, this deep crustal seismicity ceases at the longitude at which we propose slab break-off to initiate at mantle depths.

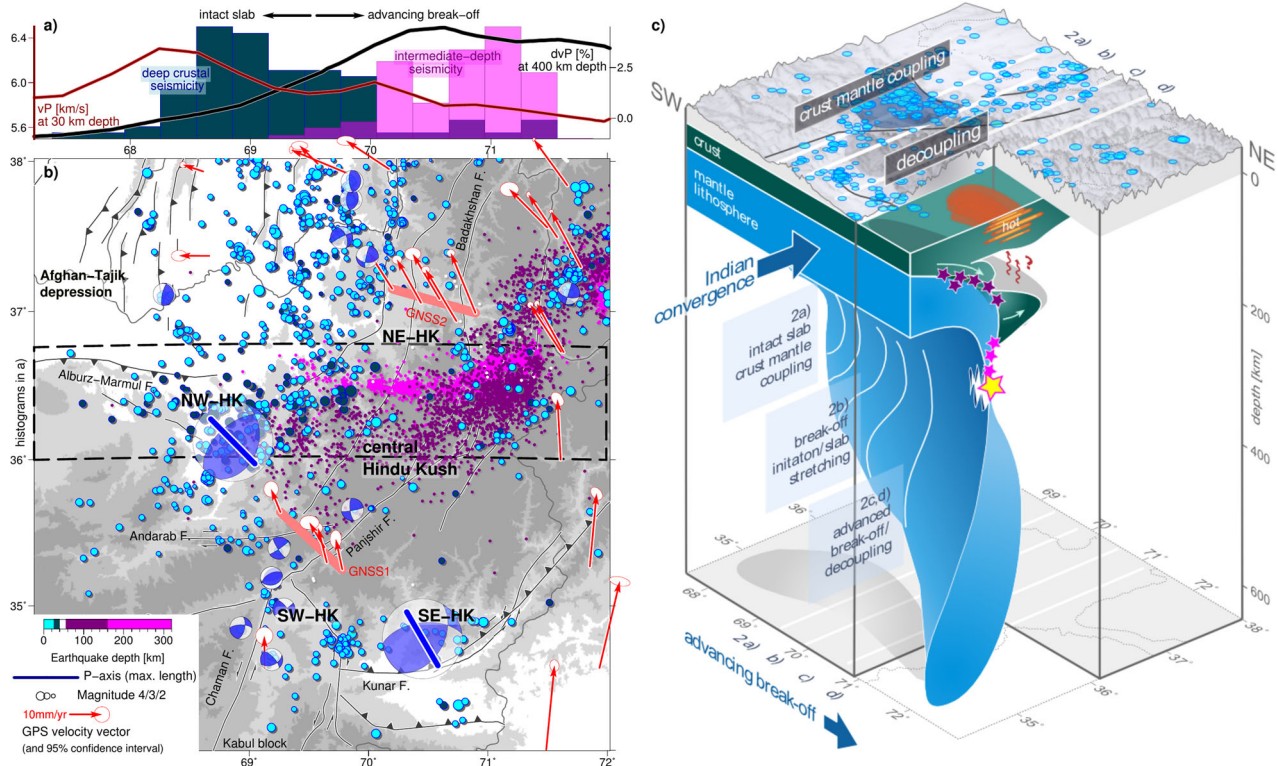

**Fig. 5 Crustal event catalogue, GNSS rates and tectonic interpretation. a** Histograms of deep crustal seismicity (25–40 km depth, 2012–2019; this study), intermediate-depth seismicity related to break-off (160–300 km depth, 2012–2019; this study and ref. [20]), and tomography results for P-wave velocity (vP) at 30 km and P-wave velocity anomaly (dvP) at 400 km depth along the longitudinal transect highlighted in (**b**). Positive dvP values east of 69.5°E indicate the presence of the stretched and tearing slab at depth. This longitudinal range is characterized by middle/lower crustal low-velocities. **b** Crustal event catalogue shallower than 40 km depth, scaled by event magnitude (see Supplementary Fig. 4 for details). Intermediate-depth seismicity (this study and ref. [20]) is plotted for orientation. The depth from 40 to ~60 km is mostly aseismic. Compression(P)-axes and focal mechanisms of crustal earthquakes from single event solutions (small beach-balls) and strain inversion (large beach balls; see Supplementary Fig. 5 for details). GNSS rates with 95% confidence ellipses relative to Asia, re-evaluated based on refs. [23,59]. GNSS1-2 highlight the locations of dense GNSS station profiles across the Hindu Kush. NE/NW/SE/SW-HK, north-east/north-west/south-east/south-west Hindu Kush. **c** Interpretation sketch illustrating the process of slab break-off and the crustal response. The slab experiences stretching and steepening during advancing break-off resulting in a greater penetration depth. Parts of the Indian crust are pulled to depth together with the slab and are buoyantly exhuming (white arrows) providing a heat input to the upper plate crust from below (wavy red arrows). Seismicity related to break-off is highlighted in pink; shallower intermediate-depth seismicity, possibly triggered by phase-transition reactions in the subducted crust, is in purple. The degree of crust-mantle coupling in the upper plate, the Hindu Kush orogen, decreases alongside the advance of slab break-off at depth. This is expressed in a change in crustal deformation style.

## Discussion

The most conspicuous features in the crustal seismicity pattern are the clusters of deeply reaching (0–30 km) earthquakes above the western end of the intermediate-depth earthquake zone (NW-HK in Figs. 2 and 5b) and the scarcity of earthquakes in the central Hindu Kush above the middle/lower crustal LVZ and above the region of most intense intermediate-depth seismicity. We suggest that these observations reflect a variable intensity and style of crustal deformation within the Hindu Kush orogen, which changes laterally, potentially dependent on or accelerated by the advancement of slab break-off at depth.

Our tomographic images show an LVZ in the mantle, which we interpret as the subducted continental crust, overlying the high-velocity slab (Fig. 2). Subduction of continental crust requires coupling between the crust of the incoming plate and its mantle lithosphere[35,45,46]. If the deeper slab sinks faster than the shallow slab, the slab must be extending, ultimately leading to the detachment of the deeper slab[5,35,46]. Break-off may be preceded by slab steepening and roll-back, which leads to the decoupling at the interface between the down-going and overriding plate that allows for asthenosphere inflow as well as the rise of crust previously attached to the sinking slab[5,35,46]. Our observations of the

central Hindu Kush slab agree well with this numerically predicted scenario. The LVZ above the mantle slab may represent crust previously attached to the sinking slab, which is now rising (schematically shown in Fig. 5c). The mantle slab shows a thinning and an overturned curvature that matches a geometry expected for north-directed subduction, followed by roll-back and break-off. In contrast, the western Hindu Kush slab does not exhibit thinning or overturning, nor a pronounced LVZ, suggesting that it is not yet notably detaching (Fig. 5c).

The question then arises how crustal deformation in the upper plate, i.e., in the Hindu Kush orogen, is related to these deep mantle processes. Numerical simulations[47,48] suggest that deformational style in an orogen strongly depends on whether the upper crust is coupled to or decoupled from the underlying mantle. Generally, deformation in the crust is coupled to the mantle motion if the orogen is cold and no decoupling horizon exists. In contrast, heating of the crust, e.g., by continuous shortening or other processes, may produce a low-viscosity layer that decouples crust from the mantle and where the crustal flow is controlled by stresses transmitted horizontally. Upper crustal motion is then only coupled to mantle motion at the flanks of the orogen. Crustal low-velocity zones in orogens, that are mostly

interpreted as regions of hot crust or partial melt, support the concept of crustal decoupling[28,49,50].

In the Hindu Kush, we observe a middle/lower crustal LVZ as well, but not along its entire extent: in the western Hindu Kush, the middle crust shows relatively high seismic velocities (Fig. 5a), indicative of cold temperatures. The GNSS rates (this study and ref. [51]) indicate an N-S shortening rate of ~10 mm/year between the station north of the Andarab fault in the central Hindu Kush and the stations showing due west displacement in the Afghan-Tajik depression, north of the Alburz-Marmul fault (Fig. 5b). Shortening across the entire Hindu Kush may be significantly larger (Fig. 5b, e.g., comparing GPS rates south of the Kunar fault and within the Tajik basin; disregarding the station in the Kabul block, that has a debated tectonic provenance[18]). This shortening appears to be accommodated by the deformation recorded by the deeply reaching thrust earthquakes (up to 30 km depth) in the NW-HK, which may define a retro-wedge. Thus, we suggest that the interior of the crust of the SW-HK remains strong and the ongoing convergence between India and the Afghan-Tajik depression is largely taken up by localized crustal shortening along the margins of the Hindu Kush, i.e., in the NW-HK and the SE-HK.

In the central Hindu Kush, low middle/lower crustal seismic velocities are observed above a domain of thickened crust (Figs. 5a and 2c, d). Therefore, the middle/lower crust probably behaves in a ductile manner causing decoupling from tectonic processes below. This explains why we do not observe deep crustal seismicity in the central Hindu Kush. Upper crustal seismicity is also reduced and clusters mainly at the southern flanks of the orogen. Further, neotectonics and geomorphic data[16,22] suggest a region of distributed deformation, which matches well with an underlying zone of ductile deformation.

The changeover from presumed upper plate coupling (high-velocity crust and crustal seismic deformation) to decoupling (low-velocity crust and absence of crustal seismic deformation) coincides spatially with the presumed onset of slab break-off at mantle depths (Fig. 5a). This correlation suggests a causal relationship, which may be provided by heat input associated with break-off from below (see the sketch in Fig. 5c). Heating through crustal shortening and thickening alone, as, e.g., suggested for the Tibetan crust[50], seems unlikely given the comparatively smaller size of the Hindu Kush orogen. Furthermore, it would not necessarily explain the variable along-strike crustal structure in the orogen. Instead, processes particular to advanced break-off may provide an additional heat source. The Hindu Kush lower crust could be heated by a partially molten subducted continental crust that is buoyantly exhuming and possibly relaminating to the hanging wall[52]. Both the drop-like LVZ in the mantle and the thick central Hindu Kush crust support such a scenario. In addition, asthenospheric inflow induced by slab roll-back and opening of a slab window maybe another heat source[5,35,46]. Lastly, partial loss of gravitational force across the rapidly extending[12,20,34] or in parts already severed slab may further contribute to the observed partitioning of hanging wall deformation. Hence, we suggest that the variable along-strike crustal deformation within the Hindu Kush orogen is influenced by processes induced by the break-off at depth.

## Methods
### Tomographic model
*Input data*. Local earthquake phase arrivals and cross-correlated teleseismic P-wave delay times measured at seismic stations from four different temporary deployment periods were integrated into this study (2008–2010, 2012–2014, 2015–2017 and 2017–2019). Together with permanent stations in the greater Pamir-Hindu Kush region, these deployments sum up to a network of 79 stations, interconnected through 19 common stations (either permanent or re-deployed campaign sites),

which were operating in at least two of the four-time periods listed above (Supplementary Table 1, Supplementary Fig. 6).

Teleseismic input data were calculated via waveform cross-correlation at two different central frequencies (1 Hz and 0.5 Hz). Events with a magnitude larger than 5.0 at epicentral distances from 30° to 90° relative to the network centre (69.94°E/37.02°N) were considered. All cross-correlation results were visually inspected, removing noisy or wrongly cross-correlated traces. Local data were assembled through the first onset picking. The P-wave first onset was picked either using a semi-automatic processing chain[53] or manually (see Supplementary Table 1 for details). No local rays with epicentral distances greater than 5° or a back-azimuthal gap larger than 210° were included in the inversion. On average, each station has 820 observations (550 local picks and 269 teleseismic delays, respectively; Supplementary Figs. 7a, b). Only stations that have both teleseismic and local picks were included in the analysis. Based on the local earthquake data, tomography models for the stations used here were published for 2008–2010 and 2012–2014 (refs. [20,28]; Supplementary Figs. 2b, c). These datasets are included in this study as groomed subsets (see Supplementary Table 1 for details). Events were de-clustered to reduce the total number of earthquakes while maintaining a representative ray-path geometry. Further, de-clustering turned out to be important to avoid introducing artefacts due to a trade-off between the relocation of closely spaced local events and the inversion for velocity anomalies.

A teleseismic tomography was calculated using data from the 2008–2010 and 2012–2014 deployments (ref. [6]; Supplementary Fig. 2d). In the model presented here, the data registered by these networks are included but input teleseismic delays were newly derived via cross-correlation and are independent of the previous study. Data associated with the 2017–2019 deployment are used here for the first time.

*Inversion procedure and synthetic travel times*. We modified the inversion scheme of ref. [26] to calculate a P-wave velocity model. Inversion and 3D ray-tracing were performed in a 715 × 890 km wide study region using approximated Born kernels[25] and a graph theory method for ray tracing (software package StingRay[54,55]). The inversion was performed based on sensitivity kernels surrounding the rays obtained in StingRay. Sensitivity kernels for teleseismic rays were calculated based on the two different filter bands used during cross-correlation for each event (1 Hz and 0.5 Hz central frequency). The local input data were considered to have 1 Hz central frequency. Regularization was achieved through damping and smoothing constraints, which are applied by minimizing the whole model norm and roughness in each iteration. Damped teleseismic event terms and station terms were included in the inversion. Poorly sampled domains in the model space were damped to prevent the creation of artificial anomalies. The quality of sampling of a specific grid node was determined through the hit-quality, a normalized measure for the number of rays crisscrossing the volume that surrounds a model node. After each inversion step, relocation of the local events was implemented by searching for the minimum root mean square travel time between observed and theoretical travel times in a cube and finer sub-cubes around the initial location. Modifications in the inversion procedure compared to ref. [26] included the extension of the ray-tracing grid above zero elevation and the input of absolute local travel times. Both favour a more accurate recovery of the absolute velocity structure in the shallow part of the study region.

We chose a two-step approach for the joint inversion, which yielded the largest data variance reduction for the entire dataset. First, a model based only on local earthquake data was calculated. This model and 3D relocated hypocentres were used as input for a joint inversion. The inversion grid (Supplementary Fig. 1) was designed in spherical coordinates, encompassing all local earthquakes and guided by the station spacing in the more densely covered regions and increased spacing in the external domains. Vertical grid spacing was increased based on ray coverage. Ray tracing was performed on a rectangular grid spanning the same domain with 5 km node spacing. Smaller ray tracing grid spacing did not significantly change the ray-tracing results. The initial 1D P-wave velocity model features regionally adequate crustal velocities including a Moho at 60 km depth[28] and standard radial Earth velocities from AK135[27] in the mantle (Supplementary Fig. 1b). Input event locations were derived by relocating the entire dataset in subsets in the 1D velocity model specified above in simulps[56], including both P and S-wave travel times and using a vP/vS ratio of 1.72 obtained from a Wadati-diagram, aiming to obtain the most accurate input locations. For both local and joint inversion, the best set of damping and smoothing parameters was determined from trade-off curves aiming to find the parameter set that yields the best compromise between data and model variance (Supplementary Fig. 8). Local and joint inversions converged after four iterations. The final data variance reduction in the local model was 72%. Data variance reduction of the final joint model, which was terminated after the seventh iteration, was 82%. Station terms are small (~<±0.15 s) and show no correlation to topography (Supplementary Fig. 7c). As for the local data only inversion, we relocated the local earthquakes after each joint inversion step, but the mean hypocentre change (0.6 km in depth) is much smaller than during the local inversion (average of 5 km depth change), indicating little change in the shallow structure. For comparison and quality control, we calculated a model based on teleseismic data only, which features the same model geometry as the joint model. In all figures shown herein, the final inversion results are interpolated onto a 5 km grid (horizontally and vertically) based on interpolation among eight surrounding grid nodes[56].

For quality control and resolution assessment, synthetic tests were performed featuring the same station-event geometry as the real data. Synthetic travel times were calculated in the StingRay software in the study region and TauP (https://www.seis.sc.edu/taup/) outside the study region using a finer ray-tracing grid (3 km) than for the inversion. Station elevations were set to zero. Gaussian noise was added to the synthetic travel times (0.1 s standard deviation (std)) and hypocentre locations were disturbed prior to inversion (4 and 2 km std vertically and horizontally, respectively).

**Local earthquake processing.** Continuous waveform data from the most recent temporary seismic network (TIPTIMON II; 4C; 2017–2019; 15 stations) and 13 nearby permanent stations were scanned using a short time average-long time average trigger. Trigger alerts were associated with possible events using a grid search approach and relocated in hypo71 based on the 1D velocity model of ref. [20]. All events shallower than 75 km in this initial event catalogue were taken as a basis for manual P and S wave phase picking. The thus derived crustal event catalogue was relocated in our 3D P-velocity model, including P and S picks (assuming a vP/vS ratio of 1.72), using the simulps and Nonlinloc software (http://alomax.free.fr/nlloc/) to obtain absolute location errors. We restrict our final earthquake catalogue to events with location errors smaller than 20 km. The final catalogue then contains 472 events shallower than 40 km depth with average horizontal and vertical location errors of 3.2 (std of 2.6) and 4.0 km (std of 2.5; Supplementary Fig. 4c). These are more events than used for the tomography, as only a selected high-quality subset was included in the inversion (see 'Methods'). Local magnitudes (ML) were calculated in Seiscomp3 (https://www.seiscomp3.org/about.html) based on the updated events. Catalogue magnitude completeness is ML = ~2.3 (Supplementary Fig. 4b).

We determined fault plane solutions from first motion polarities and S to P amplitude ratios using the HASH software[57]. P polarities were manually read from the unfiltered broadband-integrated vertical displacement seismogram. S to P ratios were obtained from the Cartesian sum of all three traces[58]. We extracted take-off angles and back-azimuths from the 3D velocity model translated into Nonlinloc travel time tables. Only solutions with 8 or more picks, back-azimuthal coverage larger than 180°, and mechanism types that remained stable upon perturbing the input data were included in the final dataset. To further account for model errors in Moho depth, only solutions, whose mechanism type remained stable in 1D velocity models with either a shallow (30 km) or deep (60 km) Moho, were accepted in the final catalogue.

Crustal fault plane solutions were further inverted to estimate the regional stress field using the software slick (https://www.usgs.gov/software/slick-package). Slick performs a linear inversion to minimize the number of rotations around an arbitrary axis necessary to rotate the input focal mechanisms to fit a uniform stress tensor. Based on the hypocentre location, we subdivided our data into two sub-regions (Supplementary Fig. 5a) aiming to fulfil the assumption of a uniform stress field. The north-west Hindu Kush (NW-HK) encompasses events between the Andarab and Alburz-Marmul faults (27 available mechanisms). The southern Hindu Kush (SW/SE-HK) includes events at the eastern edge of the Kabul block, along the Panjshir fault and events around the Kunar fault (23 available mechanisms). We accessed the robustness of the solution via a bootstrap test. The data were resampled 500 times while the selected fault slip direction is flipped in 10% of these cases. The spread of the results obtained from bootstrap inversions provides a measure of inversion robustness. Both stress tensors indicate NW-SE compression with the southern Hindu Kush featuring a component of sinistral strike-slip in addition to thrusting (Supplementary Fig. 5 and Fig. 5b).

**GNSS rates.** We reprocessed the original GNSS survey data of refs. [23,59,51] together with 24 reference stations of the International GNSS service network using the Earth Parameter and Orbit System software[60]. This software accounts for phase centre variations, ocean tide loading, ionospheric, hydrostatic and tropospheric delays. Derived positions were aligned with the International Terrestrial Reference Frame 2014[61]. We then removed outliers in the time-series by visual inspection and estimated the linear rates using a least-square approach; uncertainties were scaled by the length of the time-series. The Badakhshan GNSS surveys took place between 2015 and 2018 (GNSS profile 2 in Fig. 5b), the Panjshir surveys between 2016 and 2018 (GNSS profile 1 in Fig. 5b). Minimum displacement rates across the profiles are derived by the rate differences between the outermost profile points, assuming fault strikes of 45°E (Panjshir fault; GNSS profile 1) and 20°E (Badakhshan fault; GNSS profile 2).

## Data availability
The data that supports the findings of this study (the velocity model, local earthquake locations, the manually derived crustal earthquake catalogue, focal mechanisms and reprocessed GPS data) are attached as supplementary files to this manuscript. Raw data of the temporary networks used in this study (FDSN codes: 7B 2008–2010, https://doi.org/10.14470/2O097102; 6C 2013–2014; 5C 2012–2014, https://doi.org/10.14470/0P7567352807; 4C 2017–2019, https://doi.org/10.14470/9P7562848989) are archived at the GEOFON Data Centre and can be obtained via the GEOFON website https://geofon.gfz-potsdam.de/waveform/archive/network.php?ncode=7B&year=2008, https://geofon.gfz-potsdam.de/waveform/archive/network.php?ncode=6C&year=2013, https://geofon.gfz-potsdam.de/waveform/archive/network.php?ncode=5C&year=2012, https://geofon.gfz-potsdam.de/waveform/archive/network.php?ncode=4C&year=2017. Data of 4C (2017–2019) are restricted until 8/2023 in accordance with GEOFON data policies for temporary networks. Permanent station waveform data can be obtained through the IRIS DMC (https://ds.iris.edu/SeismiQuery/). Figures 1–5 show results derived through processing the raw seismic data. Source data are provided with this paper.

## Code availability
Software used for the derivation of the crustal earthquake catalogue, including rupture mechanisms, is open source: seiscomp3 (https://www.seiscomp.de/seiscomp3/), Nonlinloc (http://alomax.free.fr/nlloc/), simulps (e.g., http://faldersons.net/Software/Simulps/Simulps.html), HASH (https://www.usgs.gov/software/hash-12), slick (www.usgs.gov/software/slick-package), and further software specified in ref. [51]. The code for finite-frequency tomography is subject to ongoing development and research and therefore not publicly available but may be accessed upon request. The two sets of ray tracers implemented in the code are open source (www.seis.sc.edu/taup/), respectively available upon request (pages.uoregon.edu/drt/Stingray/_index.html). The Earth Parameters & Orbit System software used for GPS processing has been developed at GFZ and can be used by other institutions after signing mutual agreements.

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

## Acknowledgements

We thank the Department of Geology of Kabul University, the Afghanistan Geological Survey, the Norwegian Afghanistan Committee and the Tajik Academy of Sciences for installing and maintaining the Afghan and Tajik campaign sites, respectively. The Geophysical Instrument Pool Potsdam (GIPP) provided seismic instruments for the temporary networks; these (FDSN codes: 7B 2008–2010; 6C 2009–2010 and 2013–2014; 5C 2012–2014; 4C 2017–2019) are archived at the GEOFON Data Centre and can be obtained via the GEOFON website (https://geofon.gfz-potsdam.de/waveform/archive/index.php?type=t). Permanent station waveform data can be obtained through the IRIS DMC (https://ds.iris.edu/SeismiQuery/). This study was supported by GFZ expedition funds, and the TIPTIMON and CATENA projects, funded by the German Ministry of Science and Education (support codes 03G0809A/B and 3G0878A/B). NK was supported by an alumni scholarship from the 'UNESCO International Training Course for Seismology' at GFZ Potsdam. SK was supported by a postdoc fellowship of the 'German Academic Exchange Service (DAAD)' at the University of Minnesota. Tomographic inversion used the HPC units of the Minnesota Supercomputing Institute. Figures and

calculations used the Generic Mapping Tools (gmt.soest.hawaii.edu/), matplotlib (matplotlib.org), obspy (obspy.org) and Matlab (mathworks.com/products/matlab.html). M. Dziggel (GFZ) helped in preparing Fig. 5c and A. Gaete (GFZ) helped with first onset picking of teleseismic waveforms.

## Author contributions

S.-K.K., in interaction with M.B., calculated the tomography. N.K., in interaction with S.-K.K., W.B., and B.S. produced the crustal earthquake catalogue, focal mechanisms and crustal travel time data. S.-K.K. and M.B. contributed to the teleseismic waveform cross-correlations. N.K., S. Metzger and Z.D. contributed to the collection and processing of the GNSS data. S.K., N.K., B.S., X.Y., S. Murodkulov, L.R., and J.M. contributed to the seismological experimental design and execution, with N.K. and S. Murodkulov leading the field deployments in Afghanistan and Tajikistan. All authors contributed to the interpretation and writing of the manuscript.

## Funding

## Competing interests

The authors declare no competing interests.
