## [Peer Review File · Nature Communications]

REVIEWER COMMENTS

Reviewer #1 (Remarks to the Author):

This manuscript contains new data and comprehensive analyses and I strongly recommend for publication. I have a few comments below and the author can clarify some of my concerns.

Abstract: Why is the shallow slab less buoyant than the deeper slab?
Crust was subducted to ~150 km depth as first suggested by Roecker (1982). Do the shallow intermediate depth earthquakes occur in the subducted crust or in the mantle lithosphere?

Line 40 - After citing reference [4,5] there are no 6 and 7 and it jumps to 8-12 on line 51. Reference 14 is not cited in the text. Please check your references.

Line 47-56 – this paragraph discusses previous body-wave tomography results. It would be informative if the author would comment on the resolution of these results. This would help to establish a need for new studies.

Line 61 – Roecker (1982) indicated that the crust is subducted to at least 150 km. The author should address how far can the crust be carried into the mantle?

Line 83 – the HVZ ...and thickest (~200 km wide) in the west. What is the resolution of the HVZ?

Figure 2 – hard to see the deeper earthquakes – dark red on dark blue. Can you tune down the blue a little bit?

Figure 3 – should be iii) North-dipping instead ii) North-dipping.
iv) North-dipping cropped instead of vi) North-dipping: cropped.

Line 119 - the fourth model is cropped. What does "cropped" mean? Please explain.

What is the resolution of the depth of the slabs? The array aperture is not large enough to see down to 600 km. The authors need to address the resolution of the depth of the slab.

Line 142 – Is the subducted crust mostly the lower crust? Could the sedimentary rocks be subducted or the light upper crust needs to be detached? Some discussion on this issue would be nice.

Figure 4 – are focal mechanisms shown in blue from shallow crustal events or deeper slab events?

This brief review is by James Ni

Reviewer #2 (Remarks to the Author):

In the present manuscript the authors perform a joint tomographic inversion combining local and teleseismic earthquakes to image the crustal and mantle structure beneath the Hindu Kush region. This procedure is shown to recover with high resolution both the mantle and crustal seismic anomalies, and therefore enables a better insight into the relation between slab and crustal dynamics. This is the main contribution of this study, since other features like fast velocities imaging a northwards subducting Indian lithospheric slab, the horizontally propagating slab breakoff and subduction of crustal material up to 160-180 km were already found in former

studies, particularly in recent studies by the same first author (e.g. Kufner 2016; 2017 in EPSL). The main novelty of this study is the inference that crustal earthquakes and geodetic observations reveal crust-mantle coupling during the incipient stage of break-off, and decoupling during the advanced stage. However, in my opinion, the reasoning to support this inference is quite confusing and speculative. I further elaborate this concern in the following notes, referring to the specific parts of the manuscript that should be reassessed to be more convincing on the robustness of their conclusions.

Major comments

I found particularly problematic the Discussion section, due to the lack of convincing explanations to univocally relate the inferences of mantle-crust coupling to deep crustal seismicity and distributed deformation.

The term 'crust-mantle decoupling' used for central and eastern part of the Hindu Kush seems contradictory with the 'pull-down of crust to mantle depths' in this area (lines 202-206). This dragging down of the crust to mantle depths is only possible if they are mechanically coupled. Usually, crust and mantle are considered to be decoupled in the presence of a weak layer in the base of the lower crust, which enables independent motion of the crust and lithospheric mantle. This is usually understood as 'decoupling'. I do not follow in which sense the authors associate the presence/absence of deep crustal seismicity with crust-mantle coupling/decoupling. In subduction areas this seismicity at depths <100 km is usually related to the coupling between both plates, but not to crust-mantle coupling. I suggest the authors to better clarify this important point. Similarly, I do not understand why the decrease of deep crustal seismicity and distributed deformation is interpreted as a result of crust-mantle decoupling.

I do not understand the final sentence (lines 212-214): 'we propose that mostly aseismic and distributed, rather than localized deformation dominates in the crust during the advanced stage of slab break-off. This stands in contrast to crust-mantle coupling during incipient break-off'. The authors should provide a comprehensive mechanical interpretation for this association between slab break-off and crust-mantle decoupling, otherwise the message is weak and unclear.

Minor comments

Lines 148-149. The authors mention that part of the intermediate-depth seismicity could be due to petrological reactions associated with crustal subduction. Can the authors be more specific about the type of reaction (dehydration? Serpentinization?)

References:

Quotes to references 6 and 7 are missing in the text.

Line 54. I guess the quote should be [12-15] instead of [12,15], otherwise reference 14 is not cited in the text.

Reviewer #3 (Remarks to the Author):

The manuscript presents a complete picture of ongoing slab break-off using data recorded by several vital seismic networks covering the entire Hindu Kush region and surrounding area. The research is an important advance of significance in Earth Science, and is suitable for publication in NC. In general the quality of the figures is high, the paper is easy to read, and the discussion is interesting. However, a key point has to be clarified before acceptance.

The authors declare the gradual eastward thinning of the subducted slab mainly based on their tomographic results. If I understand correctly, the thinning is indicated by the near neutral velocity structure near 200~250 km. However, in the same range, the slab-like feature in joint inversion result (Figure 3b, iii) is also obviously weak even though the synthetic input model has a uniform and fast anomaly of at least 3% (Figure 3c, may be higher to 5% according to authors description in line 278). Similar trends exist in other synthetic end member tests (Figure 3d, i,ii and iii). Is the thinning observed in the tomographic images real or artificial due to relatively sparse data sets?

More synthetic tests are needed.

Minor comments

1. Checkerboard resolution test of tomographic model is missing.
2. Figure 3d, ii) North-dipping => iii) North-dipping
vi) => iv)

REVIEWER COMMENTS

Reviewer #1 (Remarks to the Author):

This manuscript contains new data and comprehensive analyses and I strongly recommend for publication. I have a few comments below and the author can clarify some of my concerns.

Thank you for this positive feedback.

Please find detailed replies to your comments below. Our replies are highlighted in blue, citations from the updated manuscript in green.

Abstract: Why is the shallow slab less buoyant than the deeper slab?

The deeper slab may be denser either due to the lithosphere architecture or due to mineral transformations and phase-equilibria transitions that result in denser rocks, which occur at high-pressure conditions at greater depth, such as eclogitisation of the subducted crust. The shallower slab may be more buoyant if continental crust is pulled into the subduction system.

The latter can be seen in our data. We added a sentence in the abstract accordingly:

Break-off of part of the down-going plate during continental collision occurs due to tensile stresses built-up between the deep and shallow slab, for which buoyancy is increased because of continental-crust subduction.

Crust was subducted to ~150 km depth as first suggested by Roecker (1982). Do the shallow intermediate depth earthquakes occur in the subducted crust or in the mantle lithosphere?

Most likely the bulk of the events occurs in the crust but some events may occur in the mantle lithosphere as well. This is inferred from the geometrical relationship between earthquakes and the seismic low-velocity zone but also based on other studies, which use seismics or receiver functions (Schneider et al., 2013; Mechie et al., 2012). We added a clarification in the manuscript:

In such a slab break-off and crustal-subduction scenario, the upper part of the intermediate-depth seismicity (~60-160 km; purple in Fig. 2) would mostly originate in subducted crust as the earthquakes are geometrically located mostly within the LVZ overlying the mantle slab (Fig. 2). The earthquakes could then be due to phase transitions (i.e. eclogitization), which may lead to transformational faulting^{37,38}. Isolated events may occur in the mantle lithosphere³⁹.

Line 40 - After citing reference [4,5] there are no 6 and 7 and it jumps to 8-12 on line 51.

References 6 (Schurr et al., 2014) and 7 (Mohadjer et al., 2016) are still part of the manuscript. They were cited in the Figure caption of Figure 1, which was mentioned in the text between references [4,5] and [8-12]. In the updated manuscript, these references appear with higher numbers as the figures are attached at the end of the manuscript and reference numbers are allocated automatically.

Reference 14 is not cited in the text. Please check your references.

The reference list was updated and unified. Ref. [15 (Negredo et al., 2007)] was excluded from the reference list in the updated version of the manuscript.

Line 47-56 – this paragraph discusses previous body-wave tomography results. It would be informative if the author would comment on the resolution of these results. This would help to establish a need for new studies.

We added a reference to our former teleseismic study, as an example. In this study, we explore the lower resolution limit using synthetic tests:

Ref. ⁶ for instance, used a dense local seismic network (station spacing between 30 and 60 km), but still showed that a ~30 km thick anomaly at 200 km depth would be unresolvable using teleseismic data alone.

Line 61 – Roecker (1982) indicated that the crust is subducted to at least 150 km. The author should address how far can the crust be carried into the mantle?

For the lower crust this is hard to say based on tomography. Once the mafic lower crustal minerals were transformed into eclogite-facies metamorphic assemblages, the lower crust would have similar velocities than mantle lithosphere. The situation is different for middle crust.

We referred to this issue already in the section ‘slab model’. We now extended this discussion to emphasize that possibly lower crust but unlikely middle crust is pulled to greater depths than ~150-160 km:

The LVZ, which we interpret as subducted crust terminates at ~160 km depth. Nevertheless, below ~160 km, subducted lower crust may be present, but eclogitized, making it indistinguishable from subducted mantle lithosphere ²⁷. In contrast, middle crust with andesitic or granitic composition that undergoes high- or ultrahigh-pressure metamorphic mineral transitions retains its buoyancy to ~160 km depth ^{27,35} and hence should retain its low-density, low-velocity character (e.g. velocities below ~7.5 km/s ³⁶). Therefore, middle crust is unlikely to subduct to depths greater than ~160 km.

Line 83 – the HVZ ...and thickest (~200 km wide) in the west. What is the resolution of the HVZ?

We conducted two more synthetic tests to highlight the resolution of the thickness of the slab. Figures 3d-iii & iv show a similar slab geometry to the north-dipping slab scenario but a successively thinner slab. As for the other synthetic tests of Figure 3, the input anomaly is 5% in its center, reached through a gradient. This results in a ~80 km thick slab in Fig. 3d-iii and a ~30 km thick slab in Figure 3d-iv:

We observe that both slab features can be resolved in the critical depth range from 100 to 250 km depth. The very thin slab is blurred at depths greater than 350 km. However, comparison with the real data shows that the slab-thickness is likely larger at this depth. Thus, we conclude that the thickness estimate given in Section 2 is within our resolution limit.

The conclusion from these tests is discussed in the updated version of the manuscript:

For the north-dipping scenario, we further test different slab thicknesses (Figs. 3d-iii & d-iv).

And:

The recovered anomalies in all synthetic test scenarios can be clearly distinguished from each other. However, only the north-dipping slab scenario, which includes a low-velocity zone overlying the slab to ~160 km depth (Fig. 3c) resembles the real data observations. Tests with different slab-thicknesses (Figs. 3d-iii & d-iv) illustrate that the thinning of the slab at 160-250 km depth is not a model artefact and that a slab of ~50 km thickness could still be resolved at upper mantle depths.

Figure 2 – hard to see the deeper earthquakes – dark red on dark blue. Can you tune down the blue a little bit?

We changed the color of the dark red deep earthquakes to a lighter pink instead. After playing around with the colors, this seemed to result in a clearer visual separation of anomaly and earthquakes.

The new color scale is used in all plots in which earthquakes are shown.

Figure 3 – should be iii) North-dipping instead ii) North-dipping.

iv) North-dipping cropped instead of vi) North-dipping: cropped.

We changed the figure label accordingly.

Line 119 - the fourth model is cropped. What does “cropped” mean? Please explain.

We were referring to the fact that the high velocity zone in this model is discontinuous. In the revised version, Figure 3d-iv was replaced by a more comprehensive test shown in Figure 4-iii. To be clearer, we

changed the figure label (Fig. 4-iii in the updated manuscript) to '*north-dipping slab; shallow and deep slab-gap*'. Further, we added an explanation in the figure caption:

iii) Model with similar geometry to (ii) but no mantle high velocity zone exists between 100 to 180 km depth and deeper than 360 km depth.

What is the resolution of the depth of the slabs? The array aperture is not large enough to see down to 600 km. The authors need to address the resolution of the depth of the slab.

It is true that giving 600 km as a maximum resolution depth may seem to be a very optimistic estimate. However, we are confident that our model has resolution at these depths at the spatial locations most critical for our interpretation. To illustrate this, we added a new Figure (now Fig. 4), which shows a W-E profile through the real data and two synthetic tests (with either deep or shallow terminating slab):

Furthermore, we refer to a global tomography study (Villasenor et al., 2003), in which the Hindu Kush slab also penetrates only to ~600 km depth.

We added a detailed explanation on the resolution depth in the manuscript:

The robustness of the W-E asymmetric slab-penetration depth is confirmed through the comparison shown in Figure 4: A slab that terminates at 600 km depth along the entire W-E range covered here and a slab that terminates at ~360 km depth can be distinguished from each other. Thus, we can exclude that the varying depth penetration of the slab in the real data is an artefact of velocity smearing.

We note that the intensity of the recovered anomalies as well as the depth resolution vary locally (Fig. 4c), which is attributed to varying ray coverage and event distribution. Depth resolution is up to 600 km in the west (69°E) and east (71°E), whereas it is shallower in the centre (70°E). However, comparison of the synthetic models (Figs. 4-ii & 4-iii) with the real data (Fig. 4-i) shows that all velocity anomalies introduced in Section 2.1, specifically the successive deepening of the slab, are within the resolution capacity of the tomographic inversion. Furthermore, global tomographies with larger resolution depth do not show a deeper penetration depth of the Hindu Kush slab either ^{7,8}.

Line 142 – Is the subducted crust mostly the lower crust? Could the sedimentary rocks be subducted or the light upper crust needs to be detached? Some discussion on this issue would be nice.

We think that at least part of the upper crust must be detached. We elaborated on this in an updated discussion paragraph:

The thickness of the LVZ is less than the total crustal thickness (~20-30 km vs. 65 km; ¹⁸), suggesting that only a part of the crust is pulled down. These are likely the lower and part of the middle crust as the upper crust has lowest density and would more strongly resist subduction ⁴.

Figure 4 – are focal mechanisms shown in blue from shallow crustal events or deeper slab events?

They are from the crustal events only. We added a note in the Figure caption (now Figure 5):

Compression(P)-axes and focal mechanisms of crustal earthquakes from single event solutions (small beach-balls) and strain inversion (large beach balls; see Fig. S5 for details).

Further, it is specified in the main text:

Outside the region of sparse seismicity, crustal earthquakes indicate an overall ~NW-SE compressional stress regime (Figs. 5b, S5) ...

This brief review is by James Ni

Thank you, I hope we addressed your comments adequately.

Reviewer #2 (Remarks to the Author):

Dear RVII,

Thank you for the time you spent on reviewing our manuscript. Please find detailed replies to your comments below. Our replies are highlighted in blue, citations from the updated manuscript in green.

In the present manuscript the authors perform a joint tomographic inversion combining local and teleseismic earthquakes to image the crustal and mantle structure beneath the Hindu Kush region. This procedure is shown to recover with high resolution both the mantle and crustal seismic anomalies, and therefore enables a better insight into the relation between slab and crustal dynamics. This is the main contribution of this study, since other features like fast velocities imaging a northwards subducting Indian lithospheric slab, the horizontally propagating slab breakoff and subduction of crustal material up to 160-180 km were already found in former studies, particularly in recent studies by the same first author (e.g. Kufner 2016; 2017 in EPSL).

It is true that we published before on the Hindu Kush deep lithospheric structure. However, the current study fills the missing link between deep and crustal structure. For instance, the polarity of subduction, which was still challenged after the 2017 study (Molnar & Bendick, 2019; Perry et al., 2019), is now, to our opinion, resolved much more clearly. This is to large part to new data recorded in Afghanistan published for the first time here and due to the joint tomography approach. Collecting these data was not a small feat, as may be imagined. We think this is a very important result in its own as it has strong impact on possible subduction-collision scenarios and on paleo-geographic reconstructions. To emphasize the novelty of this result, we refer to it also in the 'Tectonic scenario' and not only in the 'slab structure' section.

The mantle slab shows a thinning and an overturned curvature that matches a geometry expected for north-directed subduction, followed by roll-back and break-off. In contrast, the western Hindu Kush slab does not exhibit thinning or overturning, nor a pronounced LVZ, suggesting that it is not yet notably detaching.

The main novelty of this study is the inference that crustal earthquakes and geodetic observations reveal crust-mantle coupling during the incipient stage of break-off, and decoupling during the advanced stage. However, in my opinion, the reasoning to support this inference is quite confusing and speculative. I further elaborate this concern in the following notes, referring to the specific parts of the manuscript that should be reassessed to be more convincing on the robustness of their conclusions.

We can understand the Reviewer's concern and we reformulated the Discussion accordingly. Please find detailed comments below.

Major comments

I found particularly problematic the Discussion section, due to the lack of convincing explanations to univocally relate the inferences of mantle-crust coupling to deep crustal seismicity and distributed deformation. The term 'crust-mantle decoupling' used for central and eastern part of the Hindu Kush seems contradictory with the 'pull-down of crust to mantle depths' in this area (lines 202-206). This dragging down of the crust to mantle depths is only possible if they are mechanically coupled. Usually,

crust and mantle are considered to be decoupled in the presence of a weak layer in the base of the lower crust, which enables independent motion of the crust and lithospheric mantle. This is usually understood as 'decoupling'. I do not follow in which sense the authors associate the presence/absence of deep crustal seismicity with crust-mantle coupling/decoupling. In subduction areas this seismicity at depths <100 km is usually related to the coupling between both plates, but not to crust-mantle coupling. I suggest the authors to better clarify this important point. Similarly, I do not understand why the decrease of deep crustal seismicity and distributed deformation is interpreted as a result of crust-mantle decoupling.

We apologize for this confusion. It might partly originate from our imprecise usage of the terms coupling/decoupling, which commonly refer to the plate interface in subduction settings. Here, we were mostly referring to deformation in the upper plate, the Hindu Kush mountains. We reformulated the Discussion, carefully rephrasing the terminology and reevaluating our line of argumentation:

- The first paragraph in the '4. A tectonic scenario ..'-section summarizes the two main observations we discovered in the crustal deformation field:

The most conspicuous features in the crustal seismicity pattern are the clusters of deeply reaching (0-30 km) earthquakes above the western end of the intermediate-depth earthquake zone (NW-HK in Figs. 2 and 5b) and the scarcity of earthquakes in the central Hindu Kush above the middle/lower crustal LVZ and above the region of most intense intermediate-depth seismicity. We suggest that these observations reflect a variable intensity and style of crustal deformation within the Hindu Kush orogen, which changes laterally, potentially dependent on or accelerated by the advancement of slab break-off at depth.

- The second paragraph then refers to the deep structure and our preferred interpretation. We highlight that strong coupling between crust and mantle of the incoming plate is required to allow for the subduction of continental crust. This emphasizes the point we already made in the last paragraph of the 'Seismic imaging and slab model' section.

Our tomographic images show a LVZ in the mantle, which we interpret as subducted continental crust, overlying the high-velocity slab (Fig. 2). Subduction of continental crust requires coupling between the crust of the incoming plate and its mantle lithosphere^{34,44,45}. If the deeper slab sinks faster than the shallow slab, the slab must be extending, ultimately leading to the detachment of the deeper slab^{5,34,45}. Break-off may be preceded by slab steepening and roll-back, which leads to the decoupling at the interface between the down-going and overriding plate that allows for asthenosphere inflow as well as the rise of crust previously attached to the sinking slab^{5,34,45}. Our observations of the central Hindu Kush slab agree well with this numerically predicted scenario. The LVZ above the mantle slab may represent crust previously attached to the sinking slab, which is now rising. The mantle slab shows a thinning and an overturned curvature that matches a geometry expected for north-directed subduction, followed by roll-back and break-off. In contrast, the western Hindu Kush slab does not exhibit thinning or overturning, nor a pronounced LVZ, suggesting that it is not yet notably detaching.

- The third paragraph then discusses the consequences for crustal deformation in the upper plate, the Hindu Kush mountains. Through this separation in two paragraphs and through the addition of critical wording (like 'upper plate', 'within the Hindu Kush mountains'), it should be clear that we now refer to crustal processes in the upper plate. We note that we see a different crustal velocity structure and seismicity pattern in the western and central Hindu Kush, respectively,

and discuss in detail how such contrasting behavior in orogens is commonly explained (here different references are added). Based on this newly added background, we refer to our results and conclude that they fit well in this theoretical framework.

The question then arises how crustal deformation in the upper plate, i.e. in the Hindu Kush orogen, is related to these deep mantle processes. Numerical simulations^{46,47} suggest that deformational style in an orogen strongly depends on whether the upper crust is coupled to or decoupled from the underlying mantle. Generally, deformation in the crust is coupled to the mantle motion if the orogen is cold and no decoupling horizon exists. In contrast, heating of the crust, e.g. by continuous shortening or other processes, may produce a low-viscosity layer that decouples crust from mantle and where crustal flow is controlled by stresses transmitted horizontally. Upper crustal motion is then only coupled to mantle motion at the flanks of the orogen. Crustal low-velocity zones in orogens, that are mostly interpreted as regions of hot crust or partial melt, support the concept of crustal decoupling^{27,48,49}.

In the Hindu Kush, we observe a middle/lower crustal LVZ as well, but not along its entire extent: In the western Hindu Kush, the middle crust shows relatively high seismic velocities (Fig. 5a), indicative of cold temperatures. The GNSS rates (this study and ref.⁵⁰) indicate a N-S shortening rate of ~10 mm/yr between the station north of the Andarab fault in the central Hindu Kush and the stations showing due west displacement in the Afghan-Tajik depression, north of the Alburz-Marmul fault (Fig. 5b). Shortening across the entire Hindu Kush maybe significantly larger (Fig. 5b, e.g. comparing GPS rates south of the Kunar fault and within the Tajik basin; disregarding the station in the Kabul block, that has a debated tectonic provenance¹⁷). This shortening appears to be accommodated by the deformation recorded by the deeply reaching thrust earthquakes (up to 30 km depth) in the NW-HK, which may define a retro-wedge. Thus, we suggest that the interior of the crust of the SW-HK remains strong and the ongoing convergence between India and the Afghan-Tajik depression is largely taken up by localized crustal shortening along the margins of the Hindu Kush, i.e., in the NW-HK and the SE-HK.

In the central Hindu Kush, low middle/lower crustal seismic velocities are observed above a domain of thickened crust (Figs. 5a, 2c & d). Therefore, the middle/lower crust probably behaves in a ductile manner causing decoupling from tectonic processes below. This explains why we do not observe deep crustal seismicity in the central Hindu Kush. Upper crustal seismicity is also reduced and clusters mainly at the southern flanks of the orogen. Further, neotectonics and geomorphic data^{15,21} suggest a region of distributed deformation, which matches well with an underlying zone of ductile deformation.

- The last paragraph then bridges the gap between mantle deformation (detailed in paragraph 2) and crustal deformation (detailed in paragraph 3). Based on the correlation between results in cited literature and our findings, we clarify why we associated different domains of crustal deformation with different crust-mantle coupling.

The changeover from presumed upper plate coupling (high velocity crust and crustal seismic deformation) to decoupling (low velocity crust and absence of crustal seismic deformation) coincides spatially with the presumed onset of slab break-off at mantle depths (Fig 5a). This correlation suggests a causal relationship, which may be provided by heat input associated with break-off from below (see sketch in Fig. 5c). Heating through crustal shortening and thickening alone, as e.g. suggested for the Tibetan crust⁴⁸, seems unlikely given the comparatively smaller size of the Hindu Kush orogen. Furthermore, it would not necessarily

explain the variable along-strike crustal structure in the orogen. Instead, processes particular to advanced break-off may provide an additional heat source. The Hindu Kush lower crust could be heated by partially molten subducted continental crust that is buoyantly exhuming and possibly relaminating to the hanging wall⁵¹. Both the drop-like LVZ in the mantle and the thick central Hindu Kush crust support such a scenario. In addition, asthenospheric inflow induced by slab roll-back and opening of a slab window may be another heat source^{5,34,45}. Lastly, partial loss of gravitational force across the rapidly extending^{11,19,33} or in parts already severed slab may further contribute to the observed partitioning of hanging wall deformation. Hence, we suggest that the variable along-strike crustal deformation within the Hindu Kush orogen is influenced by processes induced by the break-off at depth.

We hope that this separation into different paragraphs helped to clarify that coupling at the interface/the incoming plate and within the upper plate are two different processes.

We also updated the abstract to emphasize the degree of certainty within the different aspects of our Discussion:

In the Hindu Kush crust, earthquakes and geodetic data show a transition from focused to distributed deformation, which we relate to a variable degree of crust-mantle coupling presumably associated with break-off at depth.

Lastly, we restructured the 'Crustal structure and deformation field section' to prepare the reader better for the discussion, which focuses on differences between western and central Hindu Kush. The section has now two paragraphs. One focuses on the central Hindu Kush and the region with largely absent crustal seismicity. The other paragraph focuses on the seismogenic part of the crust, in particular highlighting the domain of deep crustal seismicity.

I do not understand the final sentence (lines 212-214): 'we propose that mostly aseismic and distributed, rather than localized deformation dominates in the crust during the advanced stage of slab break-off. This stands in contrast to crust-mantle coupling during incipient break-off'. The authors should provide a comprehensive mechanical interpretation for this association between slab break-off and crust-mantle decoupling, otherwise the message is weak and unclear.

We removed this sentence. Instead, we elaborate in the last paragraph of the discussion on why we infer an impact from break-off on the crustal deformation (see above).

In addition, through our wording, we made clear that this is an interpretation that is not directly constrained by our data. To emphasize this, we also reformulated the last sentence of the first Discussion paragraph to:

We suggest that these observations reflect a variable intensity and style of crustal deformation within the Hindu Kush orogen, which changes laterally, potentially dependent on or accelerated by the advancement of slab break-off at depth.

Lastly, we updated the abstract accordingly:

.. **presumably** associated with break-off at depth.

Minor comments

Lines 148-149. The authors mention that part of the intermediate-depth seismicity could be due to petrological reactions associated with crustal subduction. Can the authors be more specific about the type of reaction (dehydration? Serpentinization?)

We refined the discussion on the mechanism of seismicity and refer to different published petrological models:

In such a slab break-off and crustal-subduction scenario, the upper part of the intermediate-depth seismicity (~60-160 km; purple in Fig. 2) would mostly originate in subducted crust as the earthquakes are geometrically located mostly within the LVZ overlying the mantle slab (Fig. 2). The earthquakes could then be due to phase transitions (i.e. eclogitization), which may lead to transformational faulting^{37,38}. Isolated events may occur in the mantle lithosphere³⁹. By contrast, the deepest, most vigorous seismicity occurs in high velocity material (160-300 km; pink in Fig. 2), interpreted as mantle lithosphere. These earthquakes may result directly from slab break-off: under high strain rates^{11,19,33} and relatively cold temperatures strain can localize along zones of reduced grain size due to shear-heating^{40,41}. The resulting earthquakes indicate zones of active deformation.

References:

Quotes to references 6 and 7 are missing in the text.

References 6 (Schurr et al., 2014) and 7 (Mohadjer et al., 2016) are still part of the manuscript. They were cited in the Figure caption of Figure 1, which was mentioned in the text between references [4,5] and [8-12]. In the updated manuscript, these references appear with higher numbers as the figures are attached at the end of the manuscript and reference numbers are allocated automatically.

Line 54. I guess the quote should be [12-15] instead of [12,15], otherwise reference 14 is not cited in the text.

The reference list was updated and unified: The reference was meant to be [12 (Koulakov, 2011) - (9) in the updated manuscript ,15 (Molnar & Bendick, 2019) - (11) in the updated manuscript]. Ref. [15 (Negredo et al., 2007)] was excluded from the reference list in the updated version of the manuscript.

Reviewer #3 (Remarks to the Author):

Dear RVIII,

Thank you for the time you spent on reviewing our manuscript. Please find detailed replies to your comments below. Our replies are highlighted in blue, citations from the updated manuscript in green.

The manuscript presents a complete picture of ongoing slab break-off using data recorded by several vital seismic networks covering the entire Hindu Kush region and surrounding area. The research is an important advance of significance in Earth Science, and is suitable for publication in NC. In general the quality of the figures is high, the paper is easy to read, and the discussion is interesting. However, a key point has to be clarified before acceptance.

The authors declare the gradual eastward thinning of the subducted slab mainly based on their tomographic results. If I understand correctly, the thinning is indicated by the near neutral velocity structure near 200~250 km. However, in the same range, the slab-like feature in joint inversion result (Figure 3b, iii) is also obviously weak even though the synthetic input model has a uniform and fast anomaly of at least 3% (Figure 3c, may be higher to 5% according to authors description in line 278). Similar trends exist in other synthetic end member tests (Figure 3d, i,ii and iii). Is the thinning observed in the tomographic images real or artificial due to relatively sparse data sets? More synthetic tests are needed.

Thanks for highlighting this observation. We first thought indeed that the observed artificial thinning in the synthetic tests might be an issue of ray coverage, possibly together with the ‘neutralizing’ effect of nearby positive and negative anomalies. However, further investigation showed that the thinning in the synthetic tests was produced by the input dataset.

At 200 km depth, the Hindu Kush earthquakes are most densely clustered, I.e. up to 20 events in a 5 km cube. This created a tradeoff: as we iteratively invert for velocity anomalies and then relocate the local earthquakes, the model can explain early arrivals either by making the mantle a bit faster or alternatively by moving the events a bit shallower. The damping constraint favors a smaller model norm; so the inversion "chooses" to move the event locations instead. At places where there are so many local events, the model can keep delay misfits low by fitting all the local events and it can tolerate being a little off on the teleseismic ones. In contrast, if events are less densely spaced, the iterative inversion and relocation works better.

Having detected this behavior, we further declustered our dataset in the critical depth range allowing a maximum of 6 (two from each deployment time; those with the largest number of picks) local earthquakes in 10 km cubes (see updated Table S1). Visually, the input changes little (see updated Figure 1), as still sufficient events are present at ~200 km depth, but the artificial gap in the synthetic output anomalies nearly completely disappears

Relevant part of old Figure 1 (see manuscript for full figure legend)	New dataset
---	-------------

Although the effect of modifying the input dataset on the ‘vertical-slab’ model was large, we would like to highlight that the ‘neutralizing’ effect is very localized. The observed slab-thinning in the real data affects a much broader domain.

In addition, despite this modified dataset, the inversion results based on real data change little (see updated Figure 2). This is because the local earthquakes had been located at the transition between a positive and negative velocity anomaly. Thus, the ‘neutralizing’ effect of too densely spaced earthquakes did not affect these results severely. The data variance reduction for the updated model improves slightly, however (from 80% to 82%).

Furthermore, we would like to note that the ‘progressive break-off’ postulated in the manuscript is not solely constrained from the gap in the anomaly at 200 km depth. Also, the local seismicity features different behavior for shallow and deep-intermediate depth seismicity. In addition, we observe the progressive along-strike slab-deepening. Both observations hint towards ongoing break-off.

As a reaction to RVIII's comment, we declustered our dataset and updated all models/plots, in which the input data or the final model is shown. Table S1 was updated as well.

We added a statement in the methods sections to highlight the importance of local event declustering:

Events were de-clustered to reduce the total number of earthquakes while maintaining a representative ray-path geometry. Further, de-clustering turned out to be important to avoid introducing artefacts due to a trade-off between the relocation of closely space local events and the inversion for velocity anomalies.

We note that the effect of locally neutralized anomalies is largely reduced in the new model. However, the synthetic tests still show slightly different intensity in the recovered anomalies. This is in the range expected from a tomographic inversion and likely attributed to varying ray coverage and event distribution. We commented on the different intensity of the recovered anomalies in the manuscript:

We note that the intensity of the recovered anomalies as well as the depth resolution vary locally (Fig. 4c), which is attributed to varying ray coverage and event distribution. Depth resolution is up to 600 km in the west (69°E) and east (71°E), whereas it is shallower in the centre (70°E). However, comparison of the synthetic models (Figs. 4-ii & 4-iii) with the real data (Fig. 4-i) shows that all velocity anomalies introduced in Section 2.1, specifically the successive deepening of the slab, are within the resolution capacity of the tomographic inversion.

Minor comments

1. Checkerboard resolution test of tomographic model is missing.

We did not add a checkerboard test in the initial version of the manuscript as we consider synthetic tests that are inspired by a more realistic subsurface structure as more informative. However, as RVIII specifically asked for it, we added a checkerboard test in Figure S3 together with a short description of this test:

Figure S3: Checkerboard test. Synthetic test with a varying pattern of positive and negative velocity anomalies ($\pm 5\%$). Map view dimension of the anomalies is ~ 80 km. The anomalies are separated by ~ 20 km neutral zones. The depth extent of the anomalies increases with depth from 30 km at crustal levels to up to 130 km in the mantle. In contrast to the synthetic tests in Figs. 3 & 4, no noise is added to the synthetic data and hypocentres are not disturbed. This is because the checkerboard test serves mainly as a proxy for ray coverage but does not resample realistic velocity anomalies. Results show a decrease of amplitude intensity at mantle depths. This is due to the relatively small nature of the implemented anomalies (80 km horizontal extent), which are resolved by teleseismic rays only at these depths. Synthetic tests with more realistic anomaly configurations (Figs. 3 and 4) showed a better amplitude recovery. White outline and blue/pink/purple circles represent resolution limit and local earthquakes as in Figure 2. Political boundaries are plotted in grey. The contours of the input synthetic model are highlighted in blue (positive anomaly) and red (negative anomaly).

And ‘In addition to the synthetic models of Figures 3 and 4, a checkerboard test, which is a more generic proxy for ray coverage, is included in Figure S3.’

2. Figure 3d, ii) North-dipping => iii) North-dipping

vi) => iv) Thank you. The figure labels are updated.

REVIEWERS' COMMENTS

Reviewer #2 (Remarks to the Author):

I have read carefully the rebuttal letter, and I consider that the authors have assessed correctly my comments and to those made by the other two referees. In particular, my main concern about the inferences on coupling/decoupling have now been clarified in the letter and properly addressed in the new version. I consider that the manuscript can be published without further modifications, except for a little detail. The citation to the study by Negredo et al. (2007), which has been removed in the new version, I consider it should be maintained. Yes, I am the first author of that paper, but I objectively consider that this citation is pertinent in the context of this research as it was a study about the geometry of the Hindo Kush slab constrained by the P-wave tomography by Villaseñor et al., (2003)(this citation is to an abstract of a poster presentation in the 2003 EGU meeting). Actually the citation to Negredo et al (2007) enables the reader a direct access to Villaseñor et al (2003) tomographic images for this region. An objective indicator of the relevande of the Negredo and co-authors study is that it has received nearly 200 citations.

Reviewer #3 (Remarks to the Author):

The authors have taken into account my comments and suggestions in the revised version and I have no further comment on the specific points addressed in the 1st review, except for two minor comments.

Figure 4: Considering the second model (Fig. 4iii shallow and deep slab-gap) is designed to evaluate the effect of vertical smearing on inversion results, and that there is no mantle high-velocity zone between 100 and 180 km depth, I don't understand why the authors chose the map view at 230 km depth to show it. Is it more appropriate to choose a map view between 100 and 180 km?

Figure S3: If possible, please also present the cross-section results along the sections a-d of Figure 2.

REVIEWERS' COMMENTS

Reviewer #2 (Remarks to the Author):

I have read carefully the rebuttal letter, and I consider that the authors have assessed correctly my comments and to those made by the other two referees. In particular, my main concern about the inferences on coupling/decoupling have now been clarified in the letter and properly addressed in the new version. I consider that the manuscript can be published without further modifications,

Thank you for this positive feedback.

Please find the reply to your comment below, highlighted in blue. Citations from the updated manuscript are in green.

except for a little detail. The citation to the study by Negredo et al. (2007), which has been removed in the new version, I consider it should be maintained. Yes, I am the first author of that paper, but I objectively consider that this citation is pertinent in the context of this research as it was a study about the geometry of the Hindu Kush slab constrained by the P-wave tomography by Villaseñor et al., (2003)(this citation is to an abstract of a poster presentation in the 2003 EGU meeting). Actually the citation to Negredo et al (2007) enables the reader a direct access to Villaseñor et al (2003) tomographic images for this region. An objective indicator of the relevande of the Negredo and co-authors study is that it has received nearly 200 citations.

We updated the reference list and included Negredo et. al (2007). It is indeed more sensible to include this reference instead of the EGU abstract only:

Different velocity models derived from teleseismic data exist for the Hindu Kush ⁶⁻¹⁰, consistently showing a high-velocity zone in the upper mantle. These anomalies have been interpreted either as a detaching slab of Indian origin ^{6-8,10,11}

10. Negredo, A. M., Replumaz, A., Villaseñor, A. & Guillot, S. Modeling the evolution of continental subduction processes in the Pamir-Hindu Kush region. *Earth Planet. Sci. Lett.* (2007) doi:10.1016/j.epsl.2007.04.043.

Reviewer #3 (Remarks to the Author):

The authors have taken into account my comments and suggestions in the revised version and I have no further comment on the specific points addressed in the 1st review,

We are happy that we could address your comments and suggestions.

except for two minor comments.

Please find replies to your comments below, highlighted in blue. Citations from the updated manuscript are in green.

Figure 4: Considering the second model (Fig. 4iii shallow and deep slab-gap) is designed to evaluate the effect of vertical smearing on inversion results, and that there is no mantle high-velocity zone between 100 and 180 km depth, I don't understand why the authors chose the map view at 230 km depth to show it. Is it more appropriate to choose a map view between 100 and 180 km?

Figure 4 now shows the map view section at 160 km depth. Initially we choose 230 km to be consistent with Figure 2:

Additionally, we added a comment in the shallower slab gap in the text:

comparison of the synthetic models (Figs. 4-ii & 4-iii) with the real data (Fig. 4-i) shows that all velocity anomalies introduced in Section 2.1, specifically **the tearing of the slab and the successive deepening of the slab**, are within the resolution capacity of the tomographic inversion.

Figure S3: If possible, please also present the cross-section results along the sections a-d of Figure 2.

We added two crosssections to Figure S3. These crosssections cover the region most relevant for this paper and are roughly along the profiles of Fig. 2. We did not chose the exact locations because these profiles do not run through the centres of the checkerboard anomalies.